# Preparation Process Optimization and Performance Characterization of Feed Plant Essential Oil Microcapsules

**DOI:** 10.3390/molecules27207096

**Published:** 2022-10-20

**Authors:** Qian Zhao, Yong-Sheng Gao, Fei Jin, Li-Yun Zhu

**Affiliations:** 1Aroma Engineering Technology Research and Development Center, College of Life Sciences, China Jiliang University, Hangzhou 310018, China; 2Anhui Hanfang Biotechnology Co., Ltd., Huaibei 235000, China

**Keywords:** plant essential oil, microcapsule, sharp-hole condensation method, intestinal release, feed additive

## Abstract

The exploration of safe antibiotic substitutes is one of the research hotspots in animal husbandry. Adding suitable plant essential oils into feed could improve the growth performance and immune capacity of animals. In order to make plant essential oil play a better role in feed application, sodium alginate and chitosan were used as the wall materials, and blended plant essential oils (BEO) as the core material to prepare BEO microcapsules by the sharp-hole condensation method. On the basis of single-factor experiments, the optimal preparation conditions for BEO microcapsules were obtained by response surface experiments. The physicochemical properties were characterized and analyzed by Fourier-transform infrared spectroscopy (FTIR) and field scanning electron microscope (FSEM). Meanwhile, the release mechanism was studied by simulating a gastrointestinal sustained-release experiment. The results showed that under the optimal preparation conditions, the encapsulation efficiency of BEO microcapsules could reach 80.33 ± 2.35%. FTIR and SEM analysis displayed that the microcapsules obtained had uniform color and size and a complete and compact structure. In vitro study indicated that the release amount of BEO microcapsules in the simulated intestinal fluid is higher than that in the simulated intestinal fluid, which was consistent with animal digestive and absorptive characteristics.

## 1. Introduction

The problems of bacterial drug resistance and biological enrichment caused by the long-term improper use of antibiotics in animal husbandry have seriously affected human society [1]. With the prohibition of antibiotics from animal feed in July 2020, the development of new green feed additives has become a research focus in animal husbandry in China. Plant essential oils are natural volatile liquids extracted from different aromatic parts of plants. Additionally, their major constituents could be categorized into two structural families: terpenoids and phenylpropanoids [2]. Plant essential oils would become a better alternative to traditional antibiotics in animal husbandry [3,4]. However, some plant essential oils have high viscosity and pungent odor, which not only affects the feeding of livestock and poultry but also affects the palatability of the feed [5,6]. Therefore, using environmentally friendly and safe technology to solve the problems of volatility and slow release of plant essential oils is an important step in the development of green feed additives for plant essential oils. Microencapsulation is the process of forming a functional barrier between the core material and the wall material to avoid chemical and physical reactions between the different components. Additionally, microencapsulation could release the core material under specific conditions or at a controllable rate [7]. Previous research has shown that microencapsulation could improve the oxidative stability, thermal stability, and biological activity of the core material [8]. Zhang et al. [9] prepared the β-cyclodextrin/sodium alginate double-layered microcapsules using the inclusion encapsulation method so as to improve the long-term effectiveness of essential oil. The research results of Khatibi et al. [10] showed that the microcapsules prepared with gelatin and gum Arabic improved the controlled release of oils. Microencapsulation could be roughly divided into three categories according to the different processes and principles of preparation: chemical method, physical method, and physical chemistry method [11]. The sharp-hole condensation method is one of the chemical preparation methods for microcapsules. The preparation process is to evenly disperse the core material in the wall material solution and then pour it into the coagulation liquid at a uniform speed through the sharp hole. Thus, the wall material is instantly precipitated on the surface of the core material or cross-linked to obtain microcapsule particles [12]. In view of its simple equipment, low investment, and room temperature operation, the sharp-hole condensation method has been widely used. Additionally, the microcapsules obtained have a uniform particle size, high mechanical strength, and relatively high entrapment efficiency (EE). Ning et al. [13] prepared eucalyptus essential oil microcapsules using the sharp-hole condensation method, and the EE of the microcapsules was increased to 71.17% by response surface optimization. However, owing to the simple wall material, the microcapsules prepared by single β-cyclodextrin had large pores on the surface and poor sustained release performance [14,15]. In general, the size and shape of the prepared microcapsules would be determined by the wall material and preparation method [7]. In this study, Sodium alginate (SA)/Chitosan (CS) was used as the composite wall material, and blended plant essential oils (BEO) were used as the core material. Then, the mixed emulsion of BEO and SA was dropped uniformly into the mixed coagulant liquid containing CS and CaCl_2_ through a 0.7 mm needle by the sharp-hole condensation method. After filtration and collection, the obtained microcapsules were subjected to morphology analysis, and the EE was used as the main investigation index. On the basis of five single-factor tests of SA concentration, CS concentration, CaCl_2_ concentration, pH value of the reaction system, and the amount of essential oil added, the response surface methodology was further used to determine the preparation process parameters and optimization process for the BEO microcapsules. Furthermore, the physicochemical properties of the microcapsules were characterized and analyzed by Fourier-transform infrared spectroscopy (FTIR) and field scanning electron microscope (FSEM). Additionally, the release behavior of the microcapsules was studied through in vitro broiler experiments. Therefore, microcapsule products with a good spherical shape, high EE, and ideal release behavior were obtained. The results of this study could provide an experimental reference for the development of new dietary additives.

## 2. Results

### 2.1. Single Fctor Experiments of Microcapsules

The results in Table 1, Table 2, Table 3, Table 4 and Table 5 show the relationship between the molding effect and particle size of the BEO microcapsules as a function of various factors. The concentration of SA had a significant effect on the particle size of the BEO microcapsules, while the amount of BEO added had little effect on the particle size. Microcapsule images and micrographs displayed that the color uniformity and depth of the BEO microcapsules mainly depended on the amount of BEO added. CS, CaCl_2_ concentration, and pH value also had significant effects on particle size. In addition, these factors also affected the adhesion and hardness of the microcapsules. In Figure 1, the effects of various factors on the EE and particle size of the BEO microcapsules could be observed directly. Interestingly, the influence of all five factors on the EE showed an “inverted U-shaped” trend, which first increased and then decreased. It indicated that the EE of the BEO microcapsules had an optimal value and reached the highest value under specific conditions. The single-factor experiments preliminarily determined that the SA concentration was controlled at about 2.5%, the BEO addition was controlled at 75–125 μL, the CS and CaCl_2_ concentrations were between 0.5–1.0% and 1.5–2.0%, respectively, and the pH value was maintained at about 3.6. Furthermore, the optimal preparation process for the BEO microcapsules was determined by response surface analysis.

### 2.2. Optimization of Microcapsules Preparation by Response Surface

Based on the above single-factor experimental results of BEO microcapsule preparation, four-factor and three-level experiments were designed using Box–Behnken response surface optimization analysis. The experimental design and results are shown in Table 6 and Table 7.

Table 7 was fitted by quadratic polynomial using Design-Expert10, and the regression equation of BEO microcapsule EE (Y) to pH value (A), the amount of BEO added (B), CaCl_2_ concentration (C), and CS concentration (D) was obtained as follows: Y = 81.38–0.42A + 0.19B–0.87C + 0.51D–0.032AB–1.21AC–0.24AD + 0.84BC + 0.2BD–1.13CD–0.9A^2^–1.74B^2^–3.17C^2^–3.12D^2^ (R^2^ = 0.9602).

The ANOVA analysis table in Table 8 showed that F = 24.15 in the model, *p* < 0.0001. The regression model was extremely significant. F = 2.54 in the lack of fit, *p* = 0.1915 > 0.05, it indicated that the difference was not significant. Additionally, the model had a good degree of fit. By comparing the effects of various primary factors on the EE of microcapsules, the order of influence was C > D > A > B. Among them, the primary item C (CaCl_2_ concentration) had a very significant effect on the EE of microcapsules (*p* < 0.01), and primary item A (pH value), as well as D (CS concentration), had a significant effect (*p* < 0.05), while B (the BEO addition) had no significant effect on the EE of microcapsules (*p* > 0.05). The interaction terms AC and CD had a significant effect on the EE results (*p* < 0.01), BC had a significant effect (*p* < 0.05), while AB and AD had no significant effect (*p* > 0.05). The quadratic terms A^2^, B^2^, C^2^, and D^2^ had very significant effects on the results (*p* < 0.001).

In addition, a 3D surface analysis of the interaction between AC, CD, and BC was performed, which has important implications for EE (Figure 2). Figure 2a showed that the contour lines were dense and elliptical, while the 3D surface was steep. Under the condition of a constant pH value, the EE of the microcapsules increased first and then decreased with the increase in CaCl_2_ concentration and reached the maximum when the concentration was 1.98–2.18%. Under the condition of constant CaCl_2_ concentration, the EE of the microcapsules first increased and then decreased slowly with the increase in pH value. At a pH of about 3.62, the EE of the microcapsules reached the maximum. The contour line was elliptical, and the 3D surface was steep, as shown in Figure 2b. At the same CS concentration, the EE of the microcapsules also increased first and then decreased with the increase in the CS concentration. When the CS concentration was about 0.7%, the 3D surface of the microcapsules reached the highest point, and the EE was the highest. At the same CaCl_2_ concentration, with the corresponding increase in BEO addition, the EE first showed a corresponding increase trend and then gradually decreased, reaching the optimum value around 92.56 μL to 108.57 μL (see Figure 2c).

According to the regression equation, the optimal preparation conditions for BEO microcapsules were as follows: the pH value was 3.57, the amount of BEO added was 100.87 μL, the CaCl_2_ concentration was 1.94%, and the CS concentration was 0.78%. Under these conditions, the EE was 81.52%. For the convenience of operation, the condition of pH 3.6, 100 μL BEO addition, 2.0% CaCl_2_ concentration, and 0.75% CS concentration were selected. Additionally, the validation experiment was repeated three times. The results showed that the EE of microcapsules was 80.33 ± 2.35%, and the relative error was 1.46%, which demonstrated that the regression equation fitted well and could effectively predict the actual results.

### 2.3. Analysis of FTIR Spectrums of Microcapsules

Figure 3 displays the FTIR spectrums of SA, CS, BEO, and microcapsules, the bands of different intensities at 2920 cm^−1^ could be assigned to the C-H stretching vibration on saturated carbon atoms [16]. Specifically, SA had absorption bands at 1632 cm^−1^ and 1416 cm^−1^, representing the carboxylate (-COO^−^)-induced asymmetric and symmetrical stretching vibrations, respectively, while the spectral band at 1048 cm^−1^ could be assigned to the stretching vibration of C-OH (Figure 3a). The stretching vibration of amino N-H of CS had a relatively wide band at about 3400 cm^−1^. Further, there are obvious absorption bands around 1653 cm^−1^ and 1589 cm^−1^ due to the stretching vibration of C = O and the bending vibration of N-H [17] (Figure 3b). Figure 3d shows there was a strong band at 1724 cm^−1^, which originated from the stretching vibration of fatty acid ester carbonyl C = O in BEO. Two bands at 1449 cm^−1^ and 742 cm^−1^ resulted from the bending vibrations of -CH_2_ and the C-H in-plane oscillating vibration of the lipid carbon skeleton, respectively [18]. The band at 1124 cm^−1^ was assigned to the symmetrical stretching vibration of the C-O-C bond in fatty acid ester, and the significant band at 1675 cm^−1^ could be attributed to the stretching vibration of C = C. Additionally, the absorption band with a peak point at 1626 cm^−1^ may be attributed to the C = C stretching vibration of the aromatic hydrocarbon skeleton. In addition, the absorption band at 1276 cm^−1^ possibly resulted from the stretching vibration of phenol and alcohol structure (−OH) in BEO [19].

By comparing Figure 3a–c, it could be seen that different absorption bands appear at 1633 cm^−1^, 1429 cm^−1^, and 1022 cm^−1^ in Figure 3c. They correspond to the -COO^−^ and C-OH stretching vibration bands of SA in Figure 3a, as well as the amide I band (1653 cm^−1^) and vibration of the pyranose structure (1089 cm^−1^) of CS in Figure 3b, respectively. The original bands at positions 1653 cm^−1^ and 1089 cm^−1^ in the CS spectrum were slightly shifted to the right in the microcapsule spectrum. Previous studies have shown that this displacement was caused by the ionic interaction between the positively charged amino groups and the negatively charged groups [17,20]. Additionally, due to the formation of polyelectrolyte between -NH_3_^+^ in CS and -COO^−^ on SA, the absorption band at 1022 cm^−1^ was significantly weakened. The above results could demonstrate that the microcapsule composite membrane formed by the electrostatic interaction between the wall materials was in good condition [21]. Comparing Figure 3c–e, it can be seen that the position of the absorption band in Figure 3e roughly coincides with the absorption band appearing in Figure 3c,d, but the intensity had changed, and no new absorption bands appeared [18]. These results indicated that BEO were contained in the BEO microcapsules, and the microencapsulation treatment had no effect on the chemical structure of BEO.

### 2.4. FSEM Result Analysis of BEO Microcapsules

The comparison results of Figure 4a,d showed that the BEO microcapsules had a better spherical shape with a particle size of about 0.8 μm. Compared with empty microcapsules, the surface of the BEO microcapsules had no obvious deformation and was fuller and rounder, which was consistent with the shape observed by Manjanna K [22]. It was preliminarily proved that BEO microcapsules contained core materials and had a good encapsulation structure. At 300× magnification, distinct wrinkles and ridges could be seen on the surface of the empty microcapsules (Figure 4b,e), which may be caused by the evaporation of water during drying. Compared with the BEO microcapsules, the surface thickness was more uniform, and the density was better, which indicated that the sharp-hole condensation method has a good encapsulation and protection effect on BEO, so that BEO would not volatilize due to water evaporation. By zooming in on the local area of the microcapsules (Figure 4c,f), it could be seen that the surface texture of the BEO microcapsules was more uniform and denser than that of the empty microcapsules, and there was no obvious collapse phenomenon. The results showed that the sharp-hole condensation method could achieve the sustained release of BEO.

### 2.5. Release Behaviors of BEO Microcapsules in Simulated Gastroenteric Fluid

In order to study the release behaviors of the BEO microcapsules in the gastrointestinal tract of broilers, the cumulative release rate (Q) of BEO microcapsules in Simulated gastric fluid (SGF), and Simulated intestinal fluid (SIF) were simulated using the zero-order release model, first-order release model, Higuchi model, and the Ritger–Peppas model, respectively. The fitting results are shown in Table 9.

From the R^2^ analysis of the matching results of the release behaviors of the BEO microcapsules in SGF and SIF in Table 9, it could be seen that the microcapsules in SGF and SIF were mainly released in a first-order release model, and their R^2^ were 0.98 and 0.95, respectively. The relationship between the Q of the BEO microcapsules and time in the first-order release model is shown in Figure 5, which shows a trend of rapid release at first and then gradual stability. By observing the morphology of the microcapsules at the end point of the simulated gastrointestinal fluid experiment (Figure 6), it was found that the microcapsule particles in SIF were swollen and translucent, and the solution was turbid, while the change in the microcapsule particles in SGF was relatively small, showing dark milky white, and the solution was clearer, which could be concluded that some BEO were not yet fully released. A previous study showed that alginate particles were more easily decomposed in an alkaline environment, which also confirmed the high-release effect of BEO microcapsules in intestinal fluid [23].

## 3. Discussion

Under an appropriate SA concentration, the EE of the microcapsules increased due to the formation of a network structure with suitable pore size. When the SA concentration was greater than 2.5%, the EE was decreased. It indicated that the SA concentration of 2.5% might be the limit of the encapsulation capability of the SA network structure. When the SA concentration was less than 1.5%, the microcapsules were poor, the particles had a poor spheroidizing effect, and the particles were softer and smaller. When the SA concentration and viscosity were low, SA was more easily dripped from the needle into the coagulation bath. However, the reaction of SA with Ca^2+^ and CS in the coagulation bath was insufficient, which was consistent with Chen’s results [24]. The amount of BEO added was one of the reasons affecting the shape of the microcapsules. If the amount of BEO added is too small, the color of the microcapsules becomes lighter, and there are still vacancies in the microcapsules. However, too many BEO added would lead to wasting the raw materials. Therefore, appropriately increasing the BEO content within the scope of the experimental design would be beneficial to improve the EE of the microcapsules.

As a cross-linking agent, Ca^2+^ entangles the initially independent linear molecules, which entangle each other into a dense network structure during the preparation of the microcapsules, and its concentration had an important influence on the particle size and EE of the microcapsules [12]. When the CaCl_2_ concentration was about 2.0%, the EE reached a maximum value of 73.37%. When the concentration was too low or too high, the EE dropped to 65.32%. The formation of the microcapsules depended on the degree of the cross-linking of Ca^2+^ and SA. Among them, the microcapsules formed by low Ca^2+^ had a thin gel layer, poor cross-linking degree, and tailing phenomenon. However, too high Ca^2+^ content tended to form a dense gel layer, which was detrimental to the strength and EE of the microcapsules [12,24]. The study by Wang et al. [25] showed that too high (>3.0%) or too low (<0.5%) CaCl_2_ would cause a tailing phenomenon in the prepared microcapsules. However, this phenomenon did not appear obviously in this experiment, which might be related to the CS concentration in the coagulation bath. CS was one of the main components of the wall material. However, too high or too low CS concentration was not conducive to the microcapsule formation and the EE of BEO. The reason might be that the appropriate concentrations of CS and SA formed a semipermeable polyelectrolyte membrane, which caused the BEs to diffuse outward from the microcapsules. However, the excessively complex CS on the surface of the microcapsules increases the mass of the microcapsules and occupies the interior space, resulting in a decrease in the EE of the BEO. In addition, it has been reported that when the CS concentration was more than 1.0%, the effective collision between the wall material and the core material would be decreased; thus, the EE would be reduced [26]. As the main components of the coagulation bath, the CS concentration and CaCl_2_ concentration had a very significant effect on the EE. Too high or too low a concentration would reduce the EE of the microcapsules. It implied that a high Ca^2+^ content would occupy the -COO^−^ site of the SA and CS reaction, while a high CS concentration would cause excessive tension in the coagulation bath, hindering the complexation reaction of SA/CS and lead to poor microcapsule formation and altered EE. Under the action of electrostatic force, -COO^−^ on the SA molecular chain and NH_3_^+^ on the CS molecular chain attracted each other to form a polyelectrolyte membrane [27]. When the pH value of the reaction solution increased from 3.2 to 4.0, the EE of the BEO microcapsules increased first and then decreased. The reason was that the -COO^−^ on the SA molecular chain tended to be protonated in the lower pH system [28]. This resulted in a higher concentration of the -COOH groups at the SA interface and relatively few -COO^−^ groups reacting with CS. The intermolecular electrostatic force was relatively weak, resulting in poor encapsulation ability and an uneven dispersion of the microcapsules. As the pH gradually increased, the film-forming reaction gradually increased with the addition of -COO^−^ groups. The Ca^2+^ in the reaction system would better form calcium alginate hydrogel through ion exchange so as to better encapsulate the BEO and increase microcapsule EE [29]. In the coagulation bath system, the pH and CaCl_2_ concentration were the key factors affecting the formation of the SA/CS polyelectrolyte membrane. By comparing the changes in the response surface, it was found that the effect of increasing the CaCl_2_ concentration on the EE was better than that of increasing the pH value. Additionally, the interaction between the two factors was the increase in the EE of the microcapsules.

The drug-release mechanism in the microcapsules mainly involved the passive diffusion of the drug itself and the physical degradation and enzymatic degradation of the microcapsules in the simulated environment. Additionally, it was affected by various factors, such as the physicochemical properties of the drug itself, wall materials, and concentration [30]. The BEO microcapsules were released into the gastrointestinal fluids in a first-order release model. In addition, the Q value of the BEO microcapsules in SIF was higher than that of SGF, which might be due to the degradation reaction of the originally tightly bound calcium alginate microcapsules under the action of chymotrypsin in an alkaline environment [31]. This resulted in the weakening of the electrostatic interaction between SA/CS, which ultimately led to a looser microcapsule structure and a better release effect [32]. Furthermore, it had been suggested that bile salts and phosphates in the intestinal fluid could compete for Ca^2+^ binding on the calcium alginate gel, which would cause the microcapsules to swell and dissolve in the intestinal fluid [33,34]. At about 45 min, the release results of BEO microcapsules in SGF reached a relatively stable state with a Q value of about 70%. At this point, the Q had not yet reached a stable value in SIF. It indicated that the microcapsules would be better released and absorbed in the intestinal tract, which corresponded to the biological characteristics of digestion and absorption.

## 4. Materials and Methods

### 4.1. Materials

The chemical agents used in experiments were as follows: BEO (cinnamon: thyme: peppermint = 2:3:1) were prepared by steam distillation in the laboratory, CS (degree of deacetylation ≥ 95%, molecular weight: 161.16, CAS number: 9012-76-4) and CA (molecular weight: 398.32, CAS number: 9005-38-3) were purchased from Shanghai McLean Biochemical Technology Co., Ltd., Shanghai, China. CaCl_2_ (CAS number: 10043-52-4) and KH_2_PO_4_(CAS number: 7778-77-0) were purchased from Xilong Scientific Co., Ltd., Shantou, China. Anhydrous ethanol (CAS number: 64-17-5), n-hexane (CAS number: 92112-69-1), and acetic acid (CAS number: 64-19-7) were purchased from Zhejiang Tengyu New Material Technology Co., Ltd., Huzhou, China. NaOH (CAS number: 1310-73-2) was purchased from Tianjin Comeo Chemical Reagent Co., Ltd., Tianjin, China. Tween-20 (CAS number: 9005-64-5) was purchased from Qingdao Yousuo Chemical Technology Co., Ltd., Qingdao, China. glycerin (CAS number: 56-81-5 was purchased from Solebao Technology Co., Ltd., Beijing, China. Pepsin (CAS number: 9001-75-6) and trypsin (CAS number: 9002-07-7) were purchased from Sinopharm Group Chemical Reagent Co., Ltd., Shanghai, China. The disposable syringe was purchased from Hangzhou Lanbao Haibo Biotechnology Co., Ltd., Hangzhou, China. The 0.45 μm oil-based microporous filter membrane was purchased from Shanghai Fengchen Biotechnology Co., Ltd., Shanghai, China.

### 4.2. Methods

#### 4.2.1. Preparation of BEO Microcapsules

The microcapsules were prepared using the sharp-hole condensation method. The preparation principle is shown in Figure 7a: the large amount of -COO^−^ on the SA molecular chain and -NH_3_^+^ on the CS molecular chain formed a polyelectrolyte membrane under the electrostatic action of the charged force, and then BEOs were wrapped in it. Additionally, the free Na^+^ in the SA solution and the Ca^2+^ in the coagulation bath solution formed a three-dimensional network gel with an “egg box” structure through ion exchange so that the BEO were uniformly dispersed among the alginates [27,35]. The microcapsules were prepared according to the method of Yostawonkul et al. [31]. The preparation process is shown in Figure 7b; first, the SA solution and CS/CaCl_2_ solution were prepared by stirring in a 60 °C water bath and left at room temperature overnight. Second, BEO, Tween-20 (70 μL), and glycerol (100 μL) were added to the SA solution (10 mL), stirred at 400 rpm for 10 min to prepare a pre-emulsion, and then the mixed emulsion was homogenized at 10,000 rpm for 10 min. Finally, the mixed emulsion of BEOs/SA was dripped from a syringe with a 9-gauge needle (0.7 mm) into the coagulation bath of CaCl_2_ and CS, stirred magnetically at 450 rpm, and continued for 30 min. After rinsing with distilled water, excess water and oil were wiped off with filter paper, and the wet BEO microcapsules were obtained by filtration. Further, the dried microcapsule particles were obtained in a blast-drying oven at 40 ℃ overnight.

#### 4.2.2. Calculation of Encapsulation Efficiency

According to Li et al.’s method [36], 0.1 mL of the BEO was diluted with anhydrous ethanol to various concentrations (0.005, 0.01, 0.015, 0.02, 0.03, 0.04, 0.05, and 0.06 μL/mL). Taking anhydrous ethanol as the control group, based on the measured absorbance at a wavelength of 220 nm was measured with a UV spectrophotometer (Shanghai Yuanyan Instrument Co., Ltd., Shanghai, China), and the regression equation of absorbance (A) and the BEO concentrations (C) were obtained, A = 35.122C + 0.0014 (R^2^ = 0.997).

Then, 0.5 g of accurately weighed BEO microcapsules were added to 20 mL of anhydrous ethanol, slowly stirred, and filtered 3 times. Then, 1.0 mL of the filtrate was passed through a 0.45 μm micropore and fixed in a 10 mL volumetric flask. Then, 2–3 mL of the solution was taken, the absorbance at UV 220 nm was measured, and the BEO concentration on the surface of the microcapsules (m_0_) was calculated according to the regression equation. After filtration, the microcapsules were added to 20 mL of anhydrous ethanol for ultrasonic treatment to obtain a mixed solution, which was centrifuged at 3000 rpm for 10 min, and the encapsulated BEO could be extracted with anhydrous ethanol. The concentration of BEO encapsulated in the microcapsules (m_1_) was calculated in the same way. The EE of the BEO microcapsules was calculated using the following formula:$$\mathrm{EE}\;\%=\frac{m_{1}}{m_{0}+m_{1}}\times 100\%$$
where m_0_ = the BEO concentration on the surface of the microcapsules, and m_1_ = the concentration of BEO encapsulated in the microcapsule.

The absorbance was measured 3 times in each group, and the experiment was performed in parallel 3 times.

#### 4.2.3. Single-Factor Experiment and Response Surface Analysis Optimization

The BEO microcapsules were prepared using the Section 4.2.1 method with different factors: SA concentrations (0.75, 1.0, 1.5, 2.0, 2.5, and 3.0%), the amount of BEO added (75, 100, 125, 150, 175, and 200 μL), CaCl_2_ concentrations (1.0, 1.5, 2.0, 2.5, and 3.0%), CS concentration (0.5, 0.75, 1.0, 1.5, and 2.0%), and pH values (3.2, 3.4, 3.6, 3.8, and 4.0), and the effects of each factor on the EE of microcapsules were investigated. In order to explore the optimal preparation process for BEO microcapsules, the Box–Behnken design was adopted based on the single-factor experimental results. Using the EE of the BEO microcapsules (Y) as the response value, four independent variables, including the pH value (A), the amount of BEOs added (B), CaCl_2_ concentration (C), and the CS concentration (D) at three levels were coded as −1, 0, and +1 for statistical modeling and its constraints. All of the experiments were randomized to minimize the effects of systematic errors. The least squares method was used to perform multiple regression analysis on the experimental values, and the results were fitted to the following quadratic model [37]:$$Y=\beta_{0}+\sum_{i=1}^{4}\beta_{i}X_{i}+\sum_{i=1}^{4}\beta_{\mathrm{ii}}X_{i}^{2}+\overset{3}{\underset{i=1}{\sum }}\overset{4}{\underset{j=i+1}{\sum }}\beta_{\mathrm{ij}}X_{i}X_{j}$$
where Y is *EE*, β_0_, β_i_, β_ii_, and β_ij_ are the model constants, the constant coefficients of the linear, quadratic, and interaction terms, respectively. X_1_ and X_2_ are independent variables (A–D). In addition, the absorbance of each group was measured3 times, and the parallel experiment was performed 3 times.

#### 4.2.4. Microcapsule Morphology Analysis and Particle Size Measurement

In order to better observe the encapsulation of the essential oils, the moist microcapsule particles were placed under a light microscope (Made by Shanghai Caikang Optical Instrument Co., Ltd., Shanghai, China), and the brightness and magnification of the objective lens (25×) were adjusted to observe the microcapsule sections through the eyepiece (16×). Furthermore, the average microcapsule diameter was measured from at least three angles on the BEO microcapsules with a uniform shape after drying. The statistics for each microcapsule was over 50, and each experiment was measured 3 times.

#### 4.2.5. FTIR Analyze

According to the method of Cebi et al. [19], the infrared spectrum of the BEO was collected and analyzed using the ATR accessory. The operation was roughly as follows: 1 μL of the sample was pipetted and dropped in the sample stage, the resolution was 4 cm^−1^, the number of scans was 16 times, and the infrared scanning was carried out within the range of 4000~600 cm^−1^. The molecular structures of the wall material, core material, and microcapsules were compared and analyzed using the KBr pressing method. About 2.0 mg of the samples were ground and mixed with 200 mg of dried KBr, and after pressure preparation, the film was scanned at a 4 cm^−1^ resolution 32 times, and infrared scanning was carried out in the range of 4000~600 cm^−1^ [38].

#### 4.2.6. Microcapsule Structure Characterization by FSEM

The microcapsules with the optimal preparation process and the microcapsules without BEO were placed in an electric-heating constant-temperature drying oven at 37 °C to dry for 24 h, and then a small number of dry microcapsules were evenly spread on the conductive tape, respectively. After 30 s of gold-spraying treatment, FSEM (Made by Hitachi company, Tokyo, Japan) was used for observation under 10 kV.

#### 4.2.7. Release Behaviors in Simulated Gastrointestinal Fluids of Broilers

According to “Poultry Nutrition” edited by Guo [39], the simulated broiler glandular stomach was adjusted to pH = 4.4 and the intestinal pH = 6.4, as follows: 1.0 mol/L of dilute hydrochloric acid was adjusted to pH 4.4, and 1.0 g pepsin was added per 100 mL of solution. Finally, the SGF was obtained by filtration through a 0.2 μm microporous membrane. In the same way, we dissolved 6.8 g KH_2_PO_4_ in 500 mL of water, and the pH was adjusted to 6.35 with 0.1 mol/L NaOH. Additionally, 10 g of trypsin was added and dissolved in an appropriate amount of water. After mixing the two solutions, the SIF was obtained by adding water to 1000 mL.

The absorbance at 220 nm of BEO in the n-hexane solution at different concentrations (0.005, 0.01, 0.015, 0.02, 0.03, 0.04, and 0.05 μL/mL) was measured using a UV spectrophotometer, and the linear regression equation A = 39.332C + 0.0012 (R^2^ = 0.9992) was obtained. Then, 0.5 g of accurately weighed BEO microcapsules were placed into a 100 mL conical flask, and then 50 mL of the simulated gastrointestinal fluid was added. The simulation system temperature was set at 41 °C and oscillated slowly at 80 rpm.

The microcapsules were observed at predetermined time intervals, and 1 mL of the sample was withdrawn from the system and replaced with a fresh release medium. The content of the BEO was determined under UV at 220 nm, and the Q value was calculated using Liang Bo’s method [40].
$$Q\;\left(\%\right)=\left\{\begin{array}{c} \frac{{V\rho }_{1}}{W}\times 100\;\;\;\;\;\;\;\;\;\;\;\;\;\;\;\;\;\;\;\;\;\;\;\;\;\;\;\;,n=1\\ \frac{{V\rho }_{n}+V_{e}\left(\sum_{i=2}^{i=n}\rho_{i-1}\right)}{W}\times 100,\;\;\;n\geq 2 \end{array}\right.$$
where ρ_n_ is the concentration of BEO in the release system at the nth sampling; V is the total volume of the release system; V_e_ is the volume of each sampling; W is the total concentration of BEO in the microcapsules.

#### 4.2.8. Statistical Analysis

IBM SPSS 20 and Design Expert 10 software were used for the data analysis, and the results were expressed as mean ± standard deviation (mean ± SD), and *p* < 0.05 was considered significant. Excel 2016, Origin 2021, and Design Expert 10 were used for standard curve and drawing, single-error bars for column charts, and double-error bars for the line charts.

## 5. Conclusions

Plant essential oils have good anti-inflammatory, antioxidant, bacteriostatic, and antiviral effects and are one of the potential substitutes for antibiotics. However, the instability and solubility of essential oils in the oral cavity severely limit their applications. In this study, SA and CS were used as wall materials to prepare BEO microcapsules using the sharp-hole condensation method. The optimal preparation process for BEO microcapsules was obtained by response surface optimization analysis as follows: SA concentration of 2.5%, the amount of BEOs added of 100 μL, CS concentration of 0.75%, CaCl_2_ concentration of 2.0%, and a pH of 3.6. Under these conditions, the EE of the BEO microcapsules prepared was 80.33 ± 2.35%, and the average particle size of the dried particles was maintained at around 0.8 mm. Optical microscopy, FTIR, and SEM analysis confirmed that the prepared microcapsules, which contained BEOs and the main components, did not change and that the BEOs were evenly distributed in the wall material’s structure. The release behavior of the BEO microcapsules in the simulated gastrointestinal fluids of broilers was in accordance with the first-order release model, which indicated that the release process of the BEOs was a relatively stable and smooth process. The Q value of SIF was higher than that of SGF, indicating that the release effect of the BEO microcapsules in the intestine was better, which corresponds to the digestion and absorption characteristics of the body. In conclusion, the sharp-hole condensation method can effectively realize the encapsulation of plant essential oils, and the microcapsules have good release behavior in gastrointestinal fluid, which can better ensure the absorption and utilization of plant essential oil in animals.

## Figures and Tables

**Figure 1 molecules-27-07096-f001:**
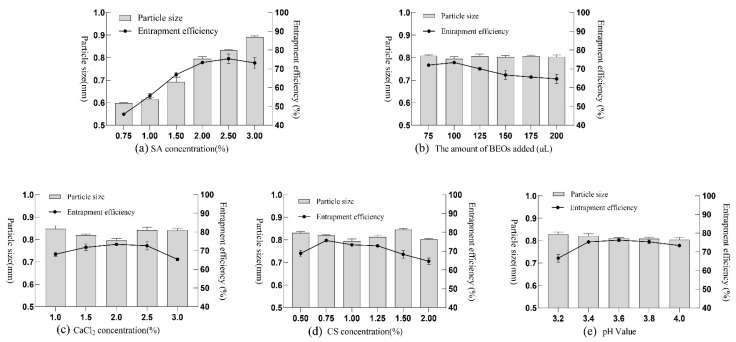
Effect of various factors on microcapsule diameter and EE. (**a**): Sodium alginate concentration; (**b**): The amount of BEO added; (**c**): CaCl_2_ concentration; (**d**): Chitosan concentration; (**e**): pH value.

**Figure 2 molecules-27-07096-f002:**
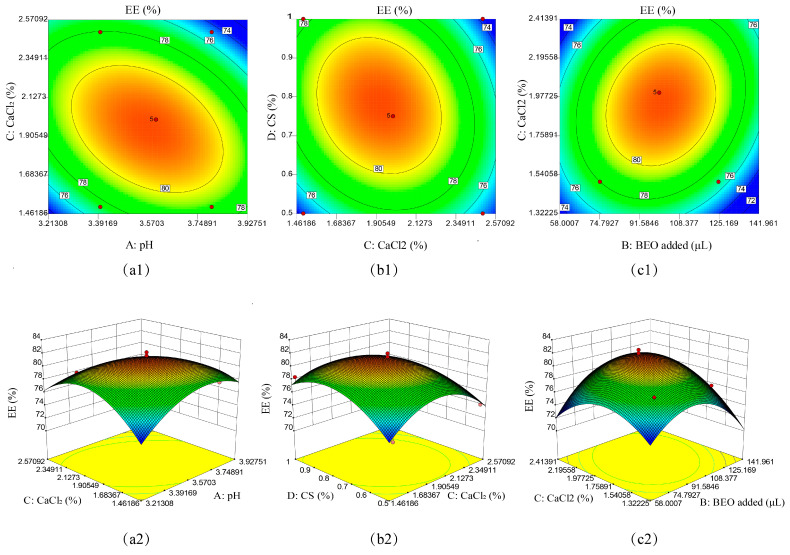
Response surface analysis diagram of the interaction of various experimental factors. The change in color from blue to red indicates a gradual increase in the interaction between the factors. The faster the change, the larger the slope, indicating the impact on the experimental results is more significant. (**a1**,**a2**): Interaction between pH Value and CaCl_2_ concentration; (**b1**,**b2**): Interaction between CaCl_2_ concentration and CS concentration; (**c1**,**c2**): Interaction between CaCl_2_ concentration and the amount of BEO added.

**Figure 3 molecules-27-07096-f003:**
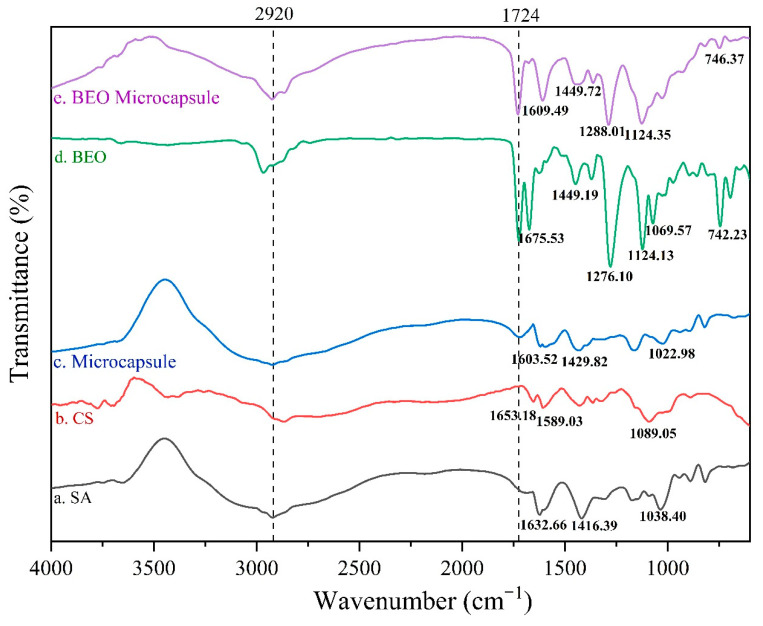
FTIR spectrums of different wall materials, BEO and microcapsules.

**Figure 4 molecules-27-07096-f004:**
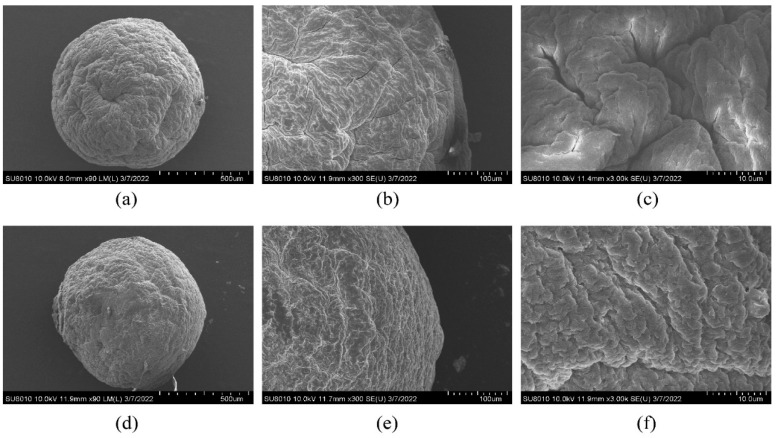
FSEM scanning micrographs of dry microcapsules. (**a**–**c**) are the structures of empty microcapsules at magnifications of 90×, 300×, and 3000×, respectively; (**d**–**f**) are the structures of BEO microcapsules at magnifications of 90×, 300×, and 3000× under the optimum preparation process.

**Figure 5 molecules-27-07096-f005:**
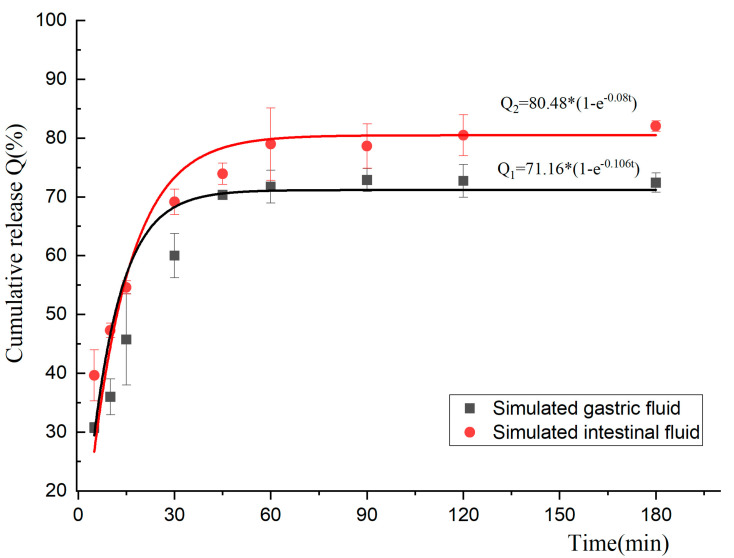
Fitting results of first-order release model of BEO microcapsules in SGF and SIF.

**Figure 6 molecules-27-07096-f006:**
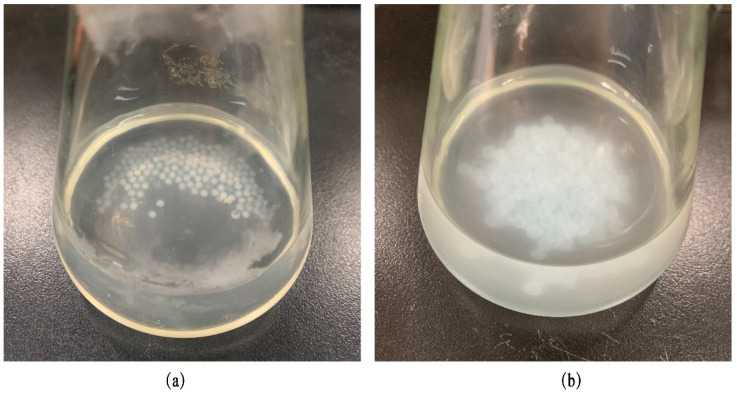
The morphology of BEO microcapsules at the end of the simulated gastrointestinal fluid experiment. (**a**)was the microcapsules in SGF and (**b**)was the microcapsules in SIF.

**Figure 7 molecules-27-07096-f007:**
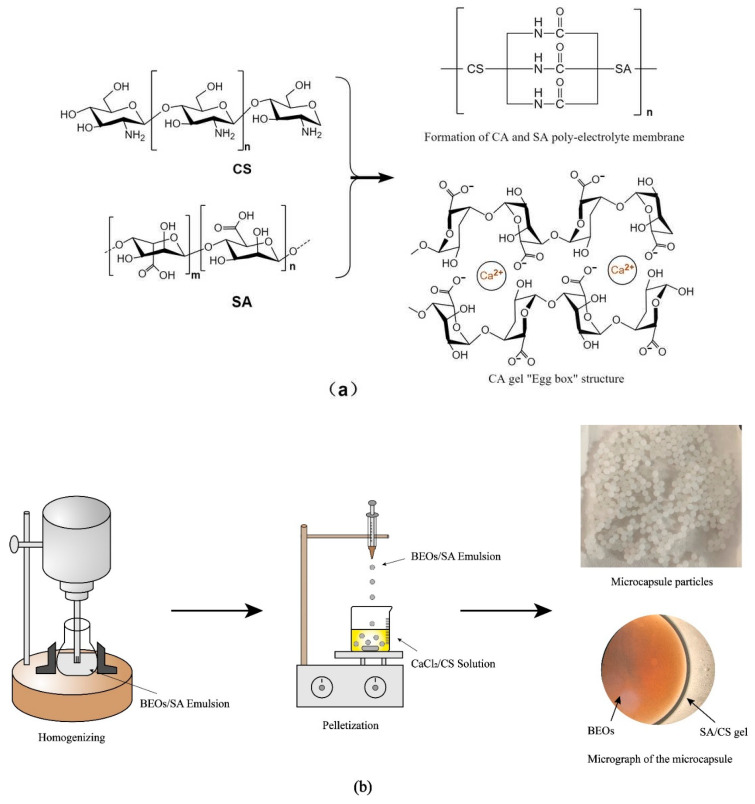
Preparation process and schematic diagram of the BEO microcapsules. (**a**) was the schematic diagram of the BEO microcapsules and (**b**) was the preparation process of the BEO microcapsules.

**Table 1 molecules-27-07096-t001:** Effect of SA Concentration on microcapsule formation.

Concentration (%)	Forming Effect	Average Dry Particle Size (mm)	Microcapsule Images	Section Micrograph(16 × 25)
0.75	Microcapsules are uneven in shape and size, with a granular flat appearance, light color, soft texture, high adhesion, and many trailing phenomena.	0.598 ± 0.002 ^e^	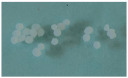	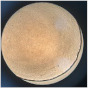
1.0	Microcapsules are relatively uniform in shape and size, with a small amount of elliptic and irregular shapes, lighter color, soft texture, high adhesion, and trailing phenomenon.	0.615 ± 0.005 ^e^	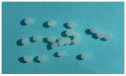	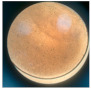
1.5	The shape of microcapsules is relatively uniform, with the existence of large or small particles, lighter color, soft texture, high adhesion, and trailing phenomenon.	0.692 ± 0.012 ^d^	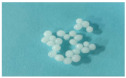	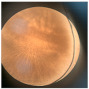
2.0	Microcapsules are uniformly in shape and size, with a good spherical shape, relatively uniform milky white in color, moderate texture, low adhesion, and no trailing phenomenon.	0.798 ± 0.005 ^c^	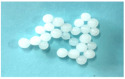	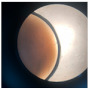
2.5	Microcapsules are uniformly in shape and size, with a good spherical shape, full milky white in color, moderate texture, low adhesion, and no trailing phenomenon.	0.833 ± 0.002 ^b^	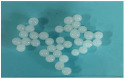	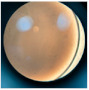
3.0	Microcapsules are relatively uniform in shape and size, with large particles, milky white color, hard texture, low adhesion, and many trailing phenomena.	0.891 ± 0.003 ^a^	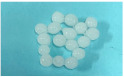	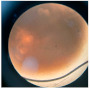

Notes: The same column of data with no letter or the same letter on the shoulder indicates a non-significant difference (*p* > 0.05), and different lowercase letters indicate a significant difference (*p* < 0.05). The following tables are the same.

**Table 2 molecules-27-07096-t002:** Effect of the amount BEOs on microcapsule formation.

The Amount of BEO Added (μL)	Forming Effect	Average Dry Particle Size (mm)	Microcapsule Images	Section Micrograph(16 × 25)
75	Microcapsules are relatively uniform in shape and size, with a good spherical shape, light and uneven milky white color, moderate texture, low adhesion, and no trailing phenomenon.	0.808 ± 0.002	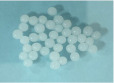	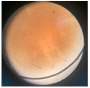
100	Microcapsules are uniformly in shape and size, with a good spherical shape, relatively uniform milky white color, moderate texture, low adhesion, and no trailing phenomenon.	0.795 ± 0.005	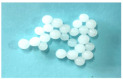	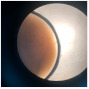
125	Microcapsules are uniformly in shape and size, with a good spherical shape, even color, moderate texture, low adhesion, and no trailing phenomenon.	0.807 ± 0.005	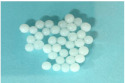	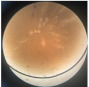
150	Microcapsules are uniformly in shape and size, with a good spherical shape, even color, moderate texture, low adhesion, and no trailing phenomenon.	0.803 ± 0.003	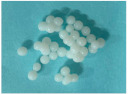	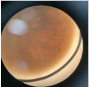
175	Microcapsules are uniform in shape and size, with a good spherical shape, deep milky white color, moderate texture, and a few have tailing phenomenon.	0.807 ± 0.002	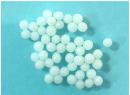	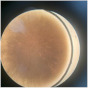
200	Microcapsules are relatively uniform in shape and size, with a few large particles, deep milky white color, hard texture, and a few have trailing phenomenon.	0.804 ± 0.004	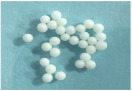	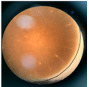

**Table 3 molecules-27-07096-t003:** Effect of CaCl_2_ concentration on microcapsule formation.

Concentration (%)	Forming Effect	Average Dry Particle Size (mm)	Microcapsule Images	Section Micrograph(16 × 25)
1.0	Microcapsules are relatively uniform in shape and size, with moderate spherical shape, light and uniform milky white color, soft texture, and moderate adhesion, and very few have tailing phenomenon.	0.847 ± 0.008 ^a^	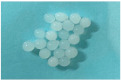	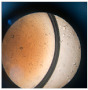
1.5	Microcapsules are relatively uniform in shape and size, with a good spherical shape, even milky white, soft texture, moderate adhesion, and no trailing phenomenon.	0.819 ± 0.002 ^b^	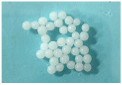	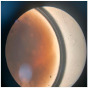
2.0	Microcapsules are uniformly in shape and size, with a good spherical shape, relatively uniform milky white color, moderate texture, low adhesion, and no trailing phenomenon.	0.795 ± 0.005 ^c^	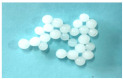	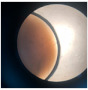
2.5	Microcapsules are uniformly in shape and size, with a good spherical shape, dark and uniform milky white color, hard texture, low adhesion, and no trailing phenomenon.	0.842 ± 0.007 ^a^	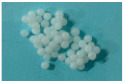	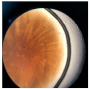
3.0	Microcapsules are uniform and round in shape, relatively uniform in size, with a good spherical shape, uneven milky white color, hard texture, low adhesion, and no trailing phenomenon.	0.841 ± 0.005 ^a^	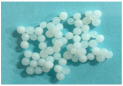	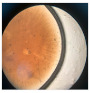

**Table 4 molecules-27-07096-t004:** Effect of CS concentration on microcapsule formation.

Concentration (%)	Forming Effect	Average Dry Particle Size (mm)	Microcapsule Images	Section Micrograph(16 × 25)
0.5	Microcapsules are uniformly in shape and size, with a good spherical shape, light milky white color, hard texture, low adhesion, and no trailing phenomenon.	0.831 ± 0.003 ^b^	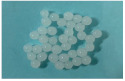	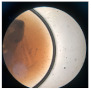
0.75	Microcapsules are relatively uniform in shape and size, with a good spherical shape, even milky white color, hard texture, low adhesion, and no trailing phenomenon.	0.822 ± 0.002 ^b,c^	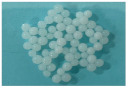	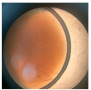
1.0	Microcapsules are uniformly in shape and size, with a good spherical shape, relatively uniform milky white color, moderate texture, low adhesion, and no trailing phenomenon.	0.795 ± 0.05 ^d^	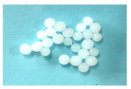	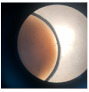
1.25	Microcapsules are smooth in shape and size, with a good spherical shape, even milky white color, hard texture, moderate adhesion, and no trailing phenomenon.	0.813 ± 0.004 ^c^	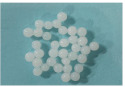	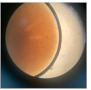
1.5	Microcapsules are uneven in shape and size, poor in a spherical shape, mostly granular, even in milky white color, moderate adhesion, moderate texture, and trailing phenomenon.	0.846 ± 0.003 ^a^	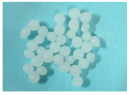	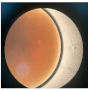
2.0	Microcapsules are uneven in shape and size, poor in spherical shape, deep milky white color, hard texture, low adhesion and have trailing phenomenon.	0.803 ± 0.002 ^d^	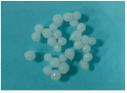	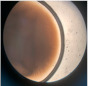

**Table 5 molecules-27-07096-t005:** Effect of pH value on microcapsule formation.

pH Value	Forming Effect	Average Dry Particle Size (mm)	Microcapsule Images	Section Micrograph(16 × 25)
3.2	Microcapsules are relatively uniform in shape, with a few large and small particles, light milky white color, moderate texture, low adhesion, and no trailing phenomenon.	0.829 ± 0.006 ^a^	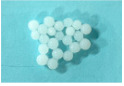	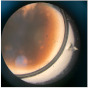
3.4	Microcapsules are uniformly in shape and size, with a good spherical shape, even milky white color, moderate texture, low adhesion and very few tailing phenomena.	0.822 ± 0.005 ^a,b^	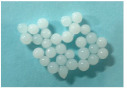	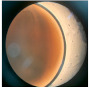
3.6	Microcapsules are uniformly in shape and size, with a good spherical shape, milky white uniform color, low adhesion, moderate texture, and no trailing phenomenon.	0.810 ± 0.002 ^b,c^	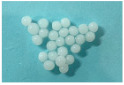	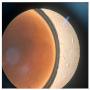
3.8	Microcapsules are uniform in shape and size, with a good spherical shape, even color, moderate texture, low adhesion, and no trailing phenomenon.	0.809 ± 0.004 ^b,c^	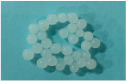	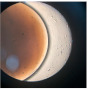
4.0	Microcapsules are uniform in shape and size, with a good spherical shape, relatively uniform milky white color, moderate texture, low adhesion, and no trailing phenomenon.	0.795 ± 0.005 ^c^	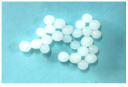	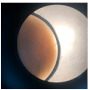

**Table 6 molecules-27-07096-t006:** Experiment design factors and coding levels.

Levels	Factors			
	A: pH Value	B: the BEO Addition/μL	C: CaCl_2_ Concentration/%	D: CS Concentration/%
−1	3.4	75	1.5	0.5
0	3.6	100	2.0	0.75
1	3.8	150	2.5	1.0

**Table 7 molecules-27-07096-t007:** Response surface experimental design and results.

No	A	B	C	D	EE/%
1	3.4	100	1.5	0.75	77.17 ± 0.28
2	3.4	100	2.0	0.5	77.26 ± 1.52
3	3.8	100	2.0	1.0	77.14 ± 1.33
4	3.6	75	1.5	0.75	78.30 ± 0.67
5	3.8	125	2.0	0.75	78.79 ± 1.26
6	3.6	100	2.0	0.75	81.22 ± 0.54
7	3.6	75	2.5	0.75	74.13 ± 0.54
8	3.6	100	1.5	1.0	78.29 ± 1.01
9	3.8	100	1.5	0.75	78.26 ± 0.10
10	3.4	125	2.0	0.75	79.59 ± 0.71
11	3.6	100	2.0	0.75	81.55 ± 0.79
12	3.6	125	1.5	0.75	77.30 ± 0.67
13	3.6	100	1.5	0.5	73.66 ± 0.38
14	3.6	75	2.0	1.0	76.31 ± 1.10
15	3.4	100	2.5	0.75	78.28 ± 0.55
16	3.6	100	2.0	0.75	82.07 ± 0.43
17	3.6	125	2.0	1.0	76.63 ± 0.73
18	3.6	100	2.0	0.75	81.15 ± 0.57
19	3.8	100	2.5	0.75	74.52 ± 0.51
20	3.8	75	2.0	0.75	78.29 ± 0.33
21	3.6	100	2.5	1.0	74.59 ± 0.70
22	3.6	75	2.0	0.5	76.32 ± 0.68
23	3.4	75	2.0	0.75	78.96 ± 0.58
24	3.8	100	2.0	0.5	77.30 ± 0.72
25	3.6	125	2.5	0.75	76.49 ± 0.56
26	3.6	100	2.5	0.5	74.49 ± 0.45
27	3.4	100	2.0	1.0	78.07 ± 0.59
28	3.6	125	2.0	0.5	75.85 ± 0.71
29	3.6	100	2.0	0.75	80.92 ± 0.71

**Table 8 molecules-27-07096-t008:** ANOVA for the response surface quadratic model.

Source	Sum of Squares	df	Mean Square	*F* Value	*p*-Value
Model	141.06	14	10.08	24.15	<0.0001
A: pH Value	2.11	1	2.11	5.05	0.0412
B: The BEO addition	0.46	1	0.46	1.09	0.3133
C: CaCl_2_ concentration	9.15	1	9.15	21.94	0.0004
D: CS concentration	3.15	1	3.15	7.56	0.0157
AB	0.0042	1	0.0042	0.010	0.9213
AC	5.88	1	5.88	14.10	0.0021
AD	0.24	1	0.24	0.56	0.4651
BC	2.82	1	2.82	6.77	0.0209
BD	0.16	1	0.16	0.37	0.5506
CD	5.13	1	5.13	12.30	0.0035
A^2^	5.29	1	5.29	12.69	0.0031
B^2^	19.58	1	19.58	46.93	<0.0001
C^2^	65.27	1	65.27	156.48	<0.0001
D^2^	63.08	1	63.08	151.23	<0.0001
Residual	5.84	14	0.42		
Lack of Fit	5.04	10	0.50	2.54	0.1915
Pure Error	0.80	4	0.20		
Cor Total	146.90	28			

**Table 9 molecules-27-07096-t009:** Release behavior fitting results of BEO microcapsules in SGF and SIF.

Models	SGF		SIF	
	Fitted Equation	R^2^	Fitted Equation	R^2^
Zero-order release	Q = 0.22t + 45.84	0.54	Q = 0.21t + 54.10	0.61
First-order release	Q = 71.16(1 − e^−0.106t^)	0.98	Q = 80.48(1 − e^−0.08t^)	0.95
Higuchi release	Q = 4.76t^0.5^ + 28.44	0.66	Q = 3.05t^0.5^ + 42.90	0.84
Ritger-Peppas release	Q = 21.71t^0.2657^	0.83	Q = 30.24t^0.2136^	0.89

## Data Availability

This study did not report any data. Informed consent was obtained from all subjects involved in the study.

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
