# Peer review of "Preparation Process Optimization and Performance Characterization of Feed Plant Essential Oil Microcapsules"

_molecules, 2022, doi:10.3390/molecules27207096_

Round 1

Reviewer 1 Report

In the manuscript submitted to me for review entitled: „Preparation process optimization and performance characterization of feed plant essential oil microcapsules the authors constructed microcapsules containing essential oils with diverse biological activities. In the process of building the microcapsules, they change various parameters to reach the most optimal composition of the capsules. The goal is that the essential oils carried by the capsules are released gradually in the body and exert their effect for a longer time.

My remarks to the authors are:

1. The abstract is a bit long. According to the requirements for drafting the manuscript should be no more than 200 words. It needs to be shortened.

2. To look at the inscription on Table 4. At the end there is an "a" that should not have a place here.

3. Wherever in the manuscript and table captions where CaCl is given the number two should be given as a subscript.

4. The caption on Table 8 "ANOVA table for regression equation" is not well worded - it should be rewritten.

Reviewer 2 Report

The authors of the manuscript reported results on microencapsulations of blended plant essential oils using the sharp-hole condensation method. While the text is well written, and the topic appears interesting to readers of Molecules, it has several general flaws that must be amended before it can be accepted for publication.

Key points:

- The introduction is misleading. While the manuscript only reports on the Preparation and characterization of essential oil microcapsules ( without conduct any antimicrobial experiments), the first paragraph of the introduction largely discussed the significance antimicrobial properties of plant essential oils. 

I suggest to re-write the intdoduction by mainly focusing on the preparation and optimazation process of microcapsules, and reducd the discussion biological activities of essential oils to be consistant with the goal of the obtained experimental results

 - Table 1 to Table 5 :  For readers convenience, it is recommanded to mention the average size of particles in the coulom of " Forming effect " and include scale bar for images in " Microcapsulaes image " and " Section micrograph '.

-  It is required to include a better quality of FTIR graph. Author should draw the graph using software such as Origin rather than paste a screen shot of the give graph .

- In line 395 :  please include comma for "  (75 100 125 150 175 and 200 μL )

- The article require an informative visual presentation (figure) in the Materials and Methods section to summarize preparation steps  of BROs microcapsules

Reviewer 3 Report

This work reports the optimization of the preparation process of plant essential oil microcapsules. The research includes the physicochemical characterization of microcapsules and in vitro analysis of microcapsules in both simulated gastric and simulated intestinal fluids. In general, the work contains relevant information for the area and is congruent with the “Molecules” journal's scope. However, the manuscript needs to improve in some aspects. The leading suggestions are described below.

1.      Table 1 to Table 5. This information is important for the optimization of the microcapsule preparation process. However, this information should be sent as a supplementary material document. The graphs included in Figure 1 already summarize the analysis generated from this optimization process.

2.      I think it would be advisable if they could include a schematic diagram showing the procedure followed for the preparation of the microcapsules, which they can display an image of the microcapsules (the one you choose according to the analysis of the optimization carried out).

3.      Page 10, lines 182, 183. Commonly, when talking about FTIR spectrum is usually referred to as absorption bands, vibration bands, etc.

4.      Figure 3. Discussion of the spectra shown in this Figure is incomplete. For example, an important piece of information is the shift to the right of the Amide I band of chitosan (approximately 1650 cm1) when it is ionically associated with another substance. Please add the corresponding discussion. I suggest documenting the information with some reference (e.g., https://doi.org/10.1002/app.47831, https://doi.org/10.1002/macp.201500034).

5.      Page 14, lines 342-343. Please include the molecular weight of chitosan and data on sodium alginate.

6.      There are multiple grammatical and writing errors in the English language.

Round 2

Reviewer 3 Report

The latest version of the manuscript is well-organized, the discussion of results has improved notably. The purpose of the work is more accurate after the suggested corrections, and it deserves to be published.